# Assessment of Micro-Hardness, Degree of Conversion, and Flexural Strength for Single-Shade Universal Resin Composites

**DOI:** 10.3390/polym14224987

**Published:** 2022-11-17

**Authors:** Pınar Yılmaz Atalı, Bengü Doğu Kaya, Aybike Manav Özen, Bilge Tarçın, Ayşe Aslı Şenol, Ezgi Tüter Bayraktar, Bora Korkut, Gülçin Bilgin Göçmen, Dilek Tağtekin, Cafer Türkmen

**Affiliations:** Department of Restorative Dentistry, Faculty of Dentistry, Marmara University, İstanbul 34854, Turkey

**Keywords:** color adjustment, elastic modulus, flexural strength, FTIR, hardness-ratio, micro-hardness, ormocer, SEM, single-shade universal composite, TCD-urethane monomer

## Abstract

Single-shade universal resin composites (SsURC) are preferred in clinical practice to reduce time for shade selection and obtain good esthetic results. In this study, the static mechanical properties of seven new SsURCs were investigated, their spectral analyzes were performed and scanning electron microscopy (SEM) evaluations were presented. Charisma Diamond One/DO, Admira Fusion x-tra/AFX, Omnichroma/OC, OptiShade/OS, Essentia Universal/EU, Zenchroma/ZC, Vittra APS Unique/VU were used in a three-point bending test to determine flexural strength (F_S_) and elastic modulus (E_M_); Vickers micro-hardness (VHN) and hardness-ratio (HR) were performed with a micro-hardness tester from top/bottom after 24-h/15-days of storage in distilled water at 37 °C (±1 °C). The degree of conversion (DC) was assessed by using Fourier transform infrared spectroscopy (FTIR). The structure of the resin matrix and filler content were assessed by SEM. Data were analyzed using IBM SPSS V23 and the R program and the significance level was taken as *p* < 0.05. The main effect of the tested SsURCs was found to be statistically significant on F_S_, E_M_, VHN, and DC values (*p* < 0.001). Bis-GMA free SsURCs (AFX, DO, VU) showed better DC and HR except for OC. All seven tested SsURCs conform to the requirements of ISO standards for dental resin composites for all tested categories.

## 1. Introduction

Direct resin composite restorations are commonly used in dental clinics, due to the improvements in the adhesive dentistry. The “Natural layering concept” was developed to provide patients’ esthetic demands and mimic natural teeth [1]. Although this technique is commonly used in dental practice, it generally requires restorative skills and long chairside time. On the other hand, recently introduced single-shade universal resin composites (SsURC) simplify the restorative procedure [2].

The SsURCs match almost all shades and thereby eliminate the shade selection step. The blending effect of the resin composite is the matching potential of restorative materials with the color of the surrounding tooth structure through reflections. When light illuminates the restorative material, it diffuses in the surface of the filler particles and scatters in multiple directions. The blending effect is affected by restoration size and translucency of the restorative material. Chameleon effect and color adjustment potential are the other common terms used to state the blending effect [3].

The composite material should show mechanical strength to withstand forces in high stress-bearing areas, otherwise, the forces could lead to body fracture and deformation of the restoration. Flexural strength (F_S_), which is one of the mechanical properties, is tested and the behavior of the material against complex stresses combining shear, compressive and tensile stresses is examined. F_S_ can be correlated with clinical wear and this correlation has been proved by the clinical and laboratory outcomes of the study [4]. Differences in F_S_ and elastic modulus (E_M_) between resin composite materials can be ascribed to the type and size of the filler as well as the filler content [5]. The procedure for measurement of 3-point bending F_S_ is determined by ISO 4049 [6].

The resin composite has three main components: resin matrix, inorganic filler particles, and silane coupling agent. Composite also has pigments, inhibitors, special additives, and initiators. With the effect of the initiation system, the cross-linking reaction begins, and the carbon-carbon double bonds are converted into carbon-carbon single bonds to form a polymer. The percentage of polymerizable double bonds converted to single bonds is indicated by the degree of conversion (DC) [7]. The final DC of current dental composites ranges from 50 to 80%, indicating that double bonds do not react in the range of 20 to 50% [8]. A high DC is associated with high polymerization shrinkage, while a low DC is associated with low mechanical properties, low color stability, and low biocompatibility [9]. In order to achieve the ideal composite, improvements in composite resins have focused on obtaining low polymerization shrinkage and high conversion degrees, therefore the modifications have been made in both organic and inorganic matrices. One of the innovative examples is the ORMOCERs (organically modified ceramic) produced with inorganic-organic hybrid co-polymer technology [10]. As an alternative to Bisphenol A-glycidyl methacrylate (Bis-GMA) and Bisphenol-A (BPA)-related monomers, in Admira Fusion x-tra, ormocer; in Charisma Diamond One, Urethane Dimethacrylate (UDMA), Triethylene glycol dimethacrylate (TEGDMA) and bis-(acryloyloxymethyl) tricycle-[5.2.1.02.6] decane (TCD-DI-HEA); in Omnichroma and Vittra APS, UDMA and TEGDMA form the organic matrix structure. According to the manufacturer, TCD-DI-HEA is claimed to induce slow cure and combine low shrinkage with low viscosity [11]. In addition to Bis-GMA, Optishade contains TEGDMA and Ethoxylated bisphenol-A dimethacrylate (Bis-EMA); Essentia Universal contains Bis-EMA, UDMA, Bisphenol-A ethoxylate dimethacrylate (Bis-MEPP) and TEGDMA; Zenchroma contains UDMA, Bis-GMA, and Tetraethylendimethacrylate (TEMDMA) monomers [12,13,14,15,16,17,18].

Limited research and information are available on the mechanical, spectral, and structural properties of seven different SsURCs. Color adjustment potential [19,20], shade match [21], optical behavior [22], color stability [1], cytotoxicity [23], F_S_, and E_M_ [24,25] of Omnichroma; quasi-static and viscoelastic behavior [26] of Charisma Diamond and Omnichroma; shrinkage [27], color change, surface roughness [28], micro-hardness [29] and contrast ratio of Vittra APS; color stability [30], F_S_ and depth of cure [31] of Optishade; color stability, surface properties [1,32], shrinkage stress, E_M_ [33] and DC [34] of Admira fusion X-tra; F_S_, E_M_, [35], color stability [36,37,38] of Essentia have been evaluated and investigated in previous research [19,24,38].

The different chemical contents and mechanical properties of resin composites imply that many different factors influence their clinical performance [39]. SsURC, having specific color-matching mechanisms through their filler structures, can exhibit different mechanical and clinical behaviors [24]. This research is focused on in vitro testing of seven different SsURCs’ various mechanical properties to obtain data that could form the basis for insight into their clinical performance.

The aim of this study was to investigate the degree of conversion, micro-hardness, hardness-ratio, flexural strength, elastic modulus, and SEM evaluation of seven different single-shade universal resin composites. The null hypothesis to be tested is that there will be no differences in the degree of conversion, micro-hardness, flexural properties, and SEM evaluation between the tested SsURCs.

## 2. Materials and Methods

This in-vitro study evaluated seven commercially available SsURCs from different manufacturers. The brand names, manufacturers, lot numbers, types, shades, composite structure (monomers, filler composition/size), and filler w/V% loadings of the SsURCs are presented in Table 1.

### 2.1. Sample Preparation

A total of 70 disc-shaped samples with the dimensions of 8 mm × 2 mm were prepared for evaluating the DC with FTIR, Vickers micro-hardness (VHN) from the top and bottom surfaces, and SEM examination. A silicone mold was positioned over a glass slide and a mylar strip (Hawe Transparent Strip, Kerr Hawe, Bioggio, Switzerland), and filled with composite in a single increment, followed by photopolymerization with an irradiance of 1100 mW/cm^2^ (Demi-Ultra LED Curing Unit, Kerr Dental, Orange, CA, USA). All samples were polished with four-step polishing discs (Finishing Discs, Bisco, Schaumburg, IL, USA) under light pressure in wet conditions using a slow-speed handpiece. The samples were kept in distilled water in a dark bottle at 37 °C (±1 °C) for 15 days.

In order to evaluate the F_S_ and the E_M_ of the SsURCs, 140 bar-shaped samples with the dimensions of 25 mm × 2 mm × 2 mm were prepared as specified by the ISO 4049/2000 specification [8]. The mold was positioned over the glass slide and the mylar strip, and the resin composite was placed in a single increment. Another mylar strip was positioned and pressed against it with a glass slide for the removal of the excess material before polymerization. The composite samples were cured with an irradiance of 1100 mW/cm^2^ (Demi-Ultra LED Curing Unit) for 20 s in three consecutive points, producing a partial overlapping. Following the light-curing, the top surfaces of the samples were polished with 600-grit paper. Finally, the samples were stored in distilled water at 37 °C (±1 °C) for 15 days.

### 2.2. Measurement of Degree of Conversion (DC)

The DC of the tested SsURCs were recorded using FTIR (FT/IR-4000, JASCO, Tokyo, Japan) with a resolution of 4 cm^−1^, 32 scans, and a spectral range of 400 to 4000 cm^−1^. Five uncured and five cured samples were measured for each selected SsURC sample, the uncured samples were spread on potassium bromide strips and their absorbance peaks were recorded. The cured samples were ground to a fine powder using a mortar and pestle. FTIR spectra were then recorded.

After applying standard baseline techniques, the spectral range between 1575 and 1660 cm^−1^ was selected and two peaks were considered for DC calculations: 1608 cm^−1^ (internal aromatic carbon double bond, C=C) and 1634 cm^−1^ (methacrylate C=C). For AFX, based on the Ormocer technique, reference peaks at 1592 cm^−1^ (C=C) and 1634 cm^−1^ (methacrylate C=C) were considered. DC is calculated according to the following formula [34]:

Degree of Conversion %
$$DC=\left[1-\frac{\left(\frac{1634{\;\mathrm{cm}}^{-1}}{1608\;{\mathrm{cm}}^{-1\;}or\;1592{\;\mathrm{cm}}^{-1}}\right)\;cured}{\;\left(\frac{1634{\;\mathrm{cm}}^{-1}}{1608\;{\mathrm{cm}}^{-1}\;or\;1592\;{\mathrm{cm}}^{-1}}\right)\;uncured}\right]\times 100$$

### 2.3. Measurement of Vickers Micro-Hardness (VHN) and Hardness-Ratio%

A total of 35 samples (five from each brand) were used for micro-hardness evaluation. Three random Vickers indentations (load: 0.49N/g; dwell time 15 s) were performed on both the top and bottom surfaces of each SsURC sample using a testing machine (Micro Hardness Tester HMV-2, Shimadzu, Tokyo, Japan) following the storage periods for 24 h and 15 days. The average of the three measurements was calculated and recorded as the micro-hardness value for each sample surface.

Vickers micro-hardness values were calculated with the following formula:$$Hv=\frac{1.8544P}{d^{2}}$$

(*Hv*: Vickers micro-hardness, *P*: the indentation load, *d*: the length of the diagonal of the indentation)

Vickers hardness-ratio of the specimens was calculated and presented using the following formula:**Hardness-ratio (HR)** =                         (Vickers hardness of bottom surface/Vickers hardness of top surface) *×* 100

### 2.4. Three-Point Bending Test

Following the storage time (24 h and 15 days), the F_S_ and E_M_ of the samples were calculated by a three-point bending test. Samples were submitted to the three-point bending test using a universal testing machine (Shimadzu AG-X Series, Shimadzu Corp., Kyoto, Japan) with a crosshead speed of 0.75 mm/min with a 20-mm supporting span. The maximum loads were obtained and the F_S_ (σ) was calculated in Megapascals (MPa) and E_M_ in Gigapascal (GPA) using the following formulas:*σ = 3FL/(2BH^2^)*

(*F*: maximum load (N); *L*: distance between the supports (mm); *B*: width of the specimen (mm); *H*: height (mm))

The E_M_ (gPa) was determined as:*E = FL^3^/4BH^3^d*

(*F*: maximum load; *L*: distance between the supports; *B*: width of the specimen, *H*: height of the specimen, *d*: deflection (mm))

### 2.5. Scanning Electron Microscopy (SEM) Evaluation

The filler morphologies of the top surfaces of SsURCs were analyzed using SEM (Zeiss EVO-MA 10, Zeiss, Oberkochen, Germany) with an acceleration voltage of 10 kV under 1000, 5000, and 10,000 magnifications. Prior to scanning and analyzing, the samples were coated with a thin layer of gold.

### 2.6. Statistical Analysis

Data were analyzed using IBM SPSS V23 (IBM, New York, NY, USA) and the R program. Conformity to the normal distribution was evaluated using the Shapiro–Wilk test.

A two-way analysis of variance was used to compare the normally distributed F_S_ values according to the composite type and time, and multiple comparisons were examined with the Tukey test. A two-way Robust test was used using the WRS21 package to compare the composite type and E_M_ values that were not normally distributed according to time, and multiple comparisons were examined with the Bonferroni test.

One-way analysis of variance (Welch) was used to compare the normally distributed DC and F_S_ values according to the composite type, and multiple comparisons were examined with Tamhane’s T2 test.

The Kruskal–Wallis test was used to compare the E_M_ values that were not normally distributed according to the composite type. Analysis results were presented as mean ± standard deviation and median (minimum-maximum) for quantitative data. The significance level was taken as *p* < 0.05.

The Kruskal–Wallis test was used to compare the normally distributed VHN and hardness-ration values according to the groups, and multiple comparisons were examined with the Dunn test. Wilcoxon test was used to compare the data that did not show normal distribution on the 24th h and 15th day in the groups. Analysis results were presented as mean ± standard deviation and median (minimum-maximum). The significance level was set at *p* < 0.05.

## 3. Results

### 3.1. Vickers Micro-Hardness (VHN) and Hardness-Ratio (HR%)

Comparison of VHN and HR values after 24 h and 15 days of the tested materials are presented in Table 2.

For ZC, DO, VU, OS, and AFX composites, no statistical difference was detected between the VHN values at 24 h and 15 days (*p* > 0.05). However, in OC and ES, there was a significant difference between the measurements taken at the 24th h and the 15th day. While the VHN values obtained at the 24th h were 74.4 for OC and 46.6 for ES, these values were measured as 177.3 and 114.4 on the 15th day, respectively (*p* = 0.043).

There was a significant difference between the VHN values of the composites obtained at the 24th h (*p* = 0.001). VHN of DO, AFX, and VU were determined as 111.3, 98.7, and 106.5, respectively, and they were significantly higher than ES, which was measured as 46.6, no statistical difference was observed between the VHN values obtained on the 15th day (*p* = 0.064).

Considering the VHN values of the top surfaces of the samples, no statistical difference was detected between the VHN values of ZC, DO, VU, OS, and AFX samples at the 24th h and 15th day (*p* > 0.050). However, a significant difference was observed between the VHN values at the end of 24 h and 15 days in OC and ES (*p* = 0.043). While the hardness values on the top surfaces of OC and ES surfaces at 24 h were measured as 117.2 and 81.3, respectively, these values were determined as 184 for OC and 205 for ES at the end of 15 days (*p* = 0.043). Although no statistical difference was observed between the composites in the VHN measurements at the end of the 24th h (*p* = 0.164), there were significant differences between the composites on the 15th day (*p* = 0.013). The VHN values for ES (205) were found to be significantly higher than OS (132.3) and AFX (128.4).

There was no statistical difference between the bottom/top VHN ratios obtained at the 24th h and 15th day in any of the groups (*p* > 0.050). Similarly, no difference was detected in terms of the bottom/top VHN ratios between composites at the 24th h and 15th day (*p* = 0.247, *p* = 0.122, respectively).

### 3.2. Degree of Conversion (DC%)

A statistically significant difference was found between the DC values according to the composite types (*p* = 0.001) (Table 3). The DC value of OC (52.09 ± 1.71) was found to be statistically significantly different from DO (65.10 ± 1.60) and VU (67.57 ± 0.86).

### 3.3. Three-Point Bending Test

The main effect of the composite brand was found to be statistically significant on F_S_ values (*p* < 0.001) (Table 4). On the other hand, the main effect of time and the interaction of composite brand and time did not have a statistically significant effect on F_S_ values (*p* = 0.264, *p* = 0.982, respectively).

Table 5 represents the multiple comparisons of the F_S_ values of the tested composites over time intervals. The highest F_S_ value was obtained in ZC (164.18 ± 37.99) and the lowest in AFX (60.38 ± 16.54).

The main effect of the composite brand was found to be statistically significant on E_M_ values (*p* < 0.001) (Table 6). It was observed that the main effect of time and the interaction of composite brand and time did not cause a statistically significant difference in E_M_ values (*p* = 0.728, *p* = 0.2783).

Multiple comparisons of E_M_ values of the tested composites according to time intervals are presented in Table 7. ZC (6.17) showed the highest E_M_ values whereas AFX (2.33) showed the lowest.

According to composite types, no statistically significant differences were observed between the F_S_ changes (*p* = 0.983) and the E_M_ changes obtained at the 24th h and 15th day (*p* = 0.683).

### 3.4. SEM Evaluation

The following findings were observed in the SEM examination of the samples obtained from the composites used in the study (Figure 1):

ES: Pre-polymerized fillers (blue arrow), nanoclusters (orange arrow) are observed in micro-hybrid structure with fillers of various sizes. OC: In a homogeneous image, the size of supra-nano spherical fillers and the distances between them are equal. A particular filler system is observed, in which nanospheres are compatible in size with the manufacturer’s specifications (260 nm). AFX: Structure of nanosphere and microparticles (20–50 nm), silicon oxide-based hybrid fillers with ormocer structure are observed. DO: Particles embedded in the TCD matrix structure were observed, together with large and small filler structures (5 nm–20 μm). The silica fillers are dispersed in the polymer matrix and are not clearly distinguishable due to nanometer size. OS: Pre-polymerized filler, nanohybrid structure, and spherical silica zirconium particles observed. VU: Homogeneous structure and uniform nanospheres are compatible in size with the manufacturer’s specifications of 200 nm. ZC: Micro-hybrid structure was seen with silicone dioxide inorganic fillers and different filler sizes (0.005–3.0 µm).

## 4. Discussion

SsURCs are preferred in clinical practice to reduce the time for shade selection and have good esthetic as well as the cost savings realized by reducing the amount of waste associated with expired products [31]. Successful resin-based composite restorations require adequate physical, mechanical and biological properties against the erosive and abrasive oral environment [40]. Currently, almost all studies focus on the esthetic properties of the SsURCs. However, there are limited studies about the mechanical or physical properties of SsURCs; for this reason, in the current study, we focused on the DC, micro-hardness, top/bottom hardness-ratio, flexural strength, elastic modulus, and SEM evaluation of seven brands of SsURC that are available for clinical practice.

DC is a crucial factor for the success of a resin-based composite restoration [40], it influences various composite properties including mechanical properties, polymerization shrinkage and stress, biocompatibility, solubility, color stability, degradation, and water sorption [40,41]. DC of light-cured resin-based composites depends on extrinsic and intrinsic factors. Intrinsic factors are the photoinitiator system, resin (monomer type, amount), and filler composition (filler size/type and amount) and the extrinsic factors include the curing time/mode, positioning of the light curing tip, irradiance, light spectrum, and post-cure reactions [31,42,43,44]. The same light curing unit (Demi-Ultra LED Curing Unit) was used in this study for the polymerization of all composite samples prepared. The irradiance of the light curing unit was measured prior to sample polymerization and it was ensured that the intensity was to be 1100 mW/cm^2^. All composite specimens were polymerized under a glass slide and from the same distance under finger pressure. Thus, type of light curing unit, distance to composite specimen and wavelength were emitted effectively in order to eliminate the effect of variations due to light curing unit as an extrinsic factor.

DC can be measured with direct (FTIR, FTIR-ATR, and FTIR-Raman Spectroscopy) and indirect methods (micro-hardness, depth of cure, differential scanning calorimetry, differential thermal calorimetry) [9]. In the current study, the DC of the tested SsURCs was measured by FTIR spectroscopy. Two different peak points were used: C=C absorption bands at 1607 cm^−1^ and 1637 cm^−1^ were used to define the DC of SsURCs and 1588 ± 4 cm^−1^ for ormocer-based composite AFX, as suggested by Balanos Carmona et al. [45].

The mean DC values of SsURC are between 52–76%. AFX (76.09%) showed the highest DC value, which is a single-shade omni-chromatic nano-ORMOCER restorative material. Contreras et al. reported that “the ormocer molecule had alkoxysilyl silane groups that allowed the formation of an inorganic link of Si-O-Si by hydrolysis and polycondensation reactions that would improve DC”, and the DC of AFX was reported as 60.11%, which was the highest of all tested materials and higher DC than methacrylates similar to our study [34].

It is known that Bis-GMA has low monomer mobility and reactivity causing decreased DC. Bis-GMA has been partially replaced with UDMA, which has low viscosity and more flexibility to overcome the disadvantages and crosslinking density, and it was diluted with TEGDMA [41,44]. In our study, the matrix structure of both VU and OC was combined with UDMA and TEGDMA monomers; DO has UDMA, TEGDMA, and a special low shrinkage monomer TCDI-HEA, which promotes the reactivity of urethane groups. This may be the reason for the rising amount of polymer crosslinks of DO [25,46,47]. ZC, AFX, and OS do not contain UDMA monomer. According to previous studies [41,48], DC increased from 30% to 60% when Bis-GMA was replaced with TEGDMA, due to the higher monomer mobility [35,44]. TCD-DI-HEA is part of the resin formulation of only DO. In our study, AFX, VU, DO, and OC are the Bis-GMA free composites and they showed higher DC values except for OC. The reason for the lowest DC of OC is based on its different matrix and filler structure. OC showed the lowest DC (52%) in the present study, but it is still in the acceptance range (50–75%). According to our current knowledge, the DC values of bulk-fill composites are between 50–81%; pre-heated ones 67–84% and after 24 h 68–86% [8]. The mean DC values of the tested SsURCs were between 52–76%, which means adequate performance was achieved.

Other intrinsic factors that influence DC are the filler composition and translucency of the resin composite. Inorganic content being 80% and over and fillers of irregular shapes decrease the transmittance of light and mean that DC decreases [34]. According to the manufacturers, the filler amount of SsURCs tested in this study were as for; AFX: 84%, VU: 82%, EU: 81.5%, OS: 81%, OC: 79%, DO: 75%, and ZC: 75% by weight. In the present study, filler content does not affect DC of the SsURCs. Although AFX has the greatest filler content (84% by weight) of all of the tested materials, it has the highest DC value. In this study, translucency parameter has been not measured and compared. All the composites are in universal shade except OS, which is medium color and may have an effect on the translucency.

Micro-hardness and DC are related properties for the resin composites [42]. In general, micro-hardness indicates the resistance of a material to plastic deformation and wear by abrasion [49]. Micro-hardness is significantly affected by increased crosslinking by polymerization reactions [50]. Therefore, micro-hardness measurements are sensitive in detecting changes in DC [51].

Vickers hardness values of dental composites spectrum are from 30 to over 100, on the other hand for mimicking the natural tooth tissues, the minimum VHN value is expected to be 40–50 [44]. Micro-hardness is also influenced by the type, morphology, and size of the filler; increasing the filler content results in higher hardness [44,52,53]. VHN values of the top surfaces of all tested composites are between 92–141 for 24 h and 126–218 for 15 days. All the VHNs are in the acceptable range for clinical practice. EU and OC were significantly different between 24 h (92.50 and 114.16) and 15 days (251 and 218.93), respectively. Greater filler composition increases the micro-hardness values of SsURCs. Bis-GMA free and high filler content SsURCs as DO, AFX, and VU showed greater VHN values for 24 h. Moreover, they (DO, VU, AFX) have greater bottom/top values than other tested composites both after 24 h and 15 days.

The bottom/top hardness ratios of the tested materials ranged between 54–86%. AF, DO, ZC, and OC have shown a value higher than 80% (the expected minimum HR) [31,46]. This can be explained by a more sensitive photo-initiator system [47] and a higher translucency, allowing a great quantity of curing light to enter and polymerize the composite equally [42,54]. Reducing the refractive index difference between the resin matrix and filler may improve DC [42]. VU and OS showed 75% HR, which is also an acceptable value, and no significant difference was detected between AF, TO, ZC, and OC. The only low HR was detected in EU (54%) material. These properties may be associated with the organic matrix. Most of the tested SsURCs had UDMA and TEGMA, but differently from the others, DO and AFX also contain TCD-urethane and ormocer, respectively (as mentioned before). Ilie et al. mentioned that TCD-urethane is characterized by higher DC and reactivity, furthermore, it is related to increased hardness values and crosslinking [23].

Light penetration of dental resin composites is positively correlated with the bottom/top ratio. Besides this factor; filler type, volume, and material shade are related factors of light penetration [31,55]. Lighter shades and smaller particles enable more light penetration [31,56]. In the present study, only OS has a medium light color; all the other tested materials are in the universal shade. No correlation was found between the color and DC of the composites. The filler size and type of the tested materials can be classified as OC is a nano-filled composite and the particle size of OC is 260 nm supra-nano spherical filler; EU (100–400 nm) and ZC (0.005–3 µm) are the micro-hybrid composites; VU (200 nm), TO (5–20 nm), OS (5–400 nm), and AFX (20–50 nm) are the nano-hybrid composites. Nano-hybrid SsURCs have HR when compared to micro-hybrid ones after 24 h. After 15 days of polymerization all the tested materials showed higher percentage of HR. This result may be related to the higher translucency of these materials. SsURCs enable reflecting the shade of the surrounding tooth structure.

Three-point flexural testing of resin-based materials was standardized by ISO [6]. In the present study, ISO testing procedures were used to perform F_S_ and E_M_ tests [6,57]. According to ISO standards, the value of the F_S_ for resin composite materials has to be 80 MPa at least. In the present study, F_S_ values are higher than 80 MPa for all tested groups except AFX (60.38). AFX is an ormocer-based composite, the type of composite is different from other groups which used in this study and the result may be associated with the composite type. The ormocer technology is defined as organically modified ceramics. Ormocer materials consist of organic-inorganic hybrid polymers, it is created by a modified siloxane network. Conventional composites showed higher long-term clinical behaviors than first-generation ormocers and this finding is coherent with this study’s results [35].

F_S_ is a considerable test for restorative materials since it describes structural reliability. E_M_ is representing the rigidity of the restorative material and it is provided by the F_S_ test [58]. F_S_ is a relevant indicator to determine the capacity to resist chewing loads, the E_M_ guides clinicians in the selection of materials suitable for the indication [35]. The test results obtained in accordance with the ISO standard were published in the manufacturers’ technical reports as follows: AFX 132 MPa; OS 140 MPa; OC 100–120 MPa/8–10 GPa [12,14,15]. In a study by Mizutani et al., the F_S_ value was reported as 116.6 MPa and the E_M_ as 6.8 GPa [24].

Bis-GMA, one of the monomers in the organic matrix content, provides physical properties such as high molecular mass and high E_M_ [59]. In some available composites today, Bis-EMA can be used instead of Bis-GMA. In the current study, EU, ZC, and OS with Bis-GMA or Bis-EMA content exhibited higher E_M_ than AFX with ormocer organic structure, and OC with TEGDMA and UDMA. TEGDMA is added to the organic matrix formulation to control the viscosity of the material [60]. VU and DO composites with TEGDMA and UDMA content showed higher E_M_ than EU and OS. Considering the results, the monomer type is not sufficient and distinctive to reach an agreement on E_M_.

Yamamoto et al. stated that the filler content is directly related to the E_M_ of the composite [61]. Pre-polymerized fillers (PPF) are significant components of resin composites as they decrease the E_M_ [60]. Besides this, they enable achieving higher luster and polishability [62]. Although OS and EU contain PPFs, F_S_ values were not higher than other groups. According to the results of the current study, no relation was found between F_S_ and the filler content of the restorative material. Previous studies have found that F_S_ can be increased with a higher volume percentage of fillers (up to 80 vol%) [25,63]. In accordance with this research, in the present study, ZC (53 vol%) and DO (59 vol%) have shown higher F_S_ values than other groups.

Light transmittance of the restorative material can be affected by filler particle size, type, volume, and morphology compromising the mechanical properties of resin composite [62]. Spherical-shaped symmetric nano-fillers with a diameter smaller than the visible wavelength light can improve the color-matching capacity of the resin composite by producing structural color without the need for pigment addition [64,65]. SiO_2_ particles with diameters of 200–300 nm can be given as examples of these nano-fillers [66]. The microstructure (particle morphology) of dental materials is assessed by SEM, it is a strong instrument for evaluating the optical and mechanical properties of restorative materials [67]. OC has an inorganic matrix composition based on uniform-sized supra-nano spherical filler combination of silicon dioxide and of zirconium dioxide (260 nm spherical particles). VU has a filler of nanospheres of zirconia complex with an average particle size of 200 nm [13]. Spherical-shaped 20–40 nm diameter nanoparticles embedded in the ormocer matrix of AFX form the filler structure as seen in the SEM radiographs.

The null hypothesis of the study was partially rejected. SsURCs showed different micro-hardness values, HR and DC; but showed similar F_S_ and E_M_ values. Differences in resin structure (monomer type and amount) and filler fraction (type, size, and amount) gave unique properties to SsURCs and some of these properties may be traced on SEM images.

One of the limitations of the current study is that the study is an in vitro study. It is not possible for laboratory studies to reflect the intraoral conditions exactly. Additionally, SsURCs have been tested, and no other universal composites (such as multi-shade composites) or bulk-fill composites have been considered. Another limitation of the study was that SsURC samples were not subjected to aging. In this context, long-term results should be observed.

Further in vivo studies are necessary to approve the effectiveness of SsURCs from the clinical aspects such as discoloration, wear rates (erosive, abrasive, and mechanical), bacterial adhesion, plaque accumulation, and toxicity.

## 5. Conclusions

The tested SsURCs exhibited similar mechanical, spectral, and structural properties with each other. This may be due to the similarities in composition and type of the tested materials. However, the structural arrangement of the fillers and the monomer type play an important role to determine the characteristic of SsURCs.
All seven tested materials fell within the ISO requirements for dental resin composites for all tested categories.Bis-GMA free SsURCs including the ormocer/TCD monomer showed higher DC and HR.AFX showed the highest DC but the lowest F_S_ value.No correlation was found among the amount of filler particles with DC and F_S_ of SsURCs. However, micro-hardness and HR values increased with having higher filler content.SEM evaluations revealed smoother surfaces with OC due to its unique spherical and similar-sized filler particles.

## Figures and Tables

**Figure 1 polymers-14-04987-f001:**
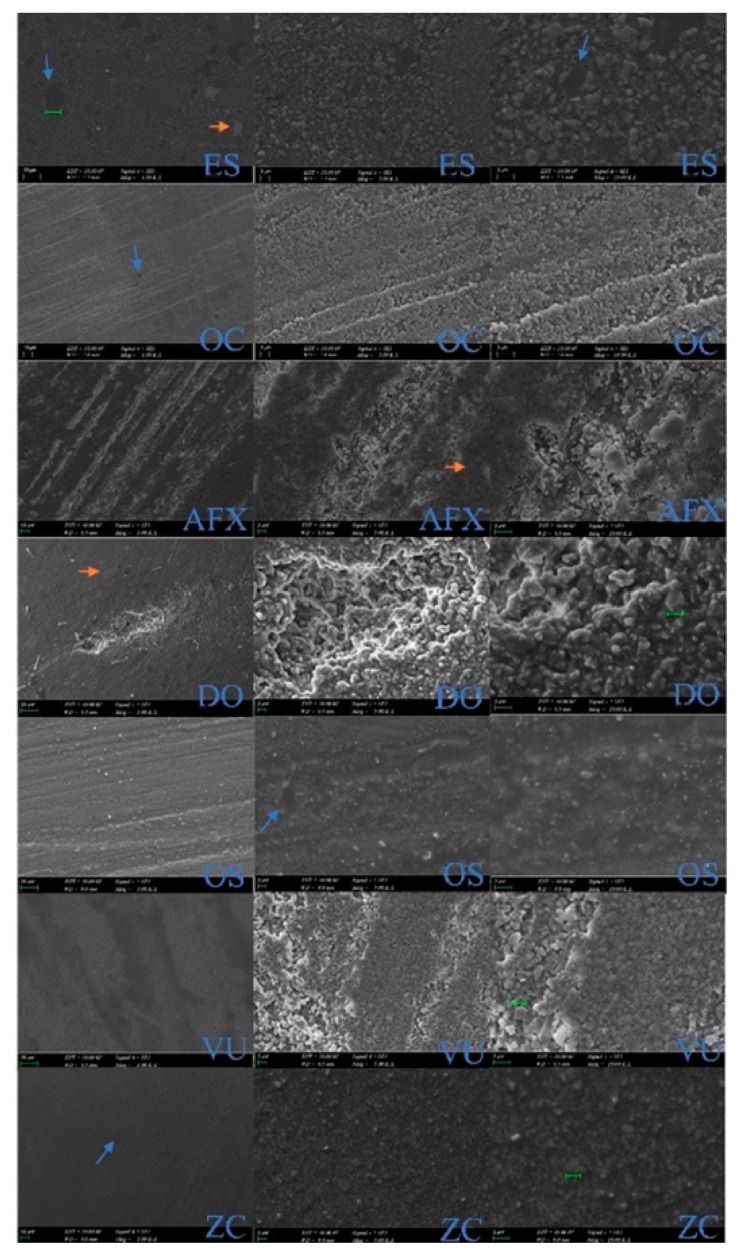
SEM images, electron backscatter diffraction mode (1000, 5000, and 10,000 magnifications, respectively).

**Table 1 polymers-14-04987-t001:** The brand names, manufacturers, lot numbers, shades, monomers, filler types, and filler loadings of the composites.

Material	Manufacturer *	Lot Number	Type	Shade	Composite Structure			Code
					Monomer	Filler Composition/Size	Filler w/V%	
Omnichroma	Tokuyama, Japan	6,00E+30	Nanofilled	Universal	UDMA TEGDMA	Uniform sized supra-nano spherical filler (260 nm spherical SiO_2_-ZrO_2_) and CF	79/68	OC
Vittra APS Unique	FGM, Brasil	21020	Nanohybrid	Universal	UDMA TEGDMA	Zirconia charge, silica (200 nm)	82/72	VU
Charisma Diamond One	Kulzer, Germany	K010021	Nanohybrid	Universal	UDMA TCD-DI-HEA TEGDMA	B2O3-F-Al2O3-SiO2, silica, TiO_2_, fluorescent pigments, metallic oxide pigments, organic pigments, 5 nm–20 μm	81/64	DO
OptiShade	Kerr Dental, USA	8242079	Nanohybrid	Medium	Bis-EMA Bis-GMA TEGDMA	PPF, BaO-Al2O3-SiO2, silica, and F_3_Yb, organic fillers Smallest primary particle size: 5 nm, Largest primary particle size: 400 nm, average particle size: 50 nm	81/64.5	OS
Admira Fusion x-tra	VOCO GmbH, Germany	2135509	Nanohybrid	Universal	ORMOCER	Silicon dioxide nanofillers (20–50 nm) and silicon oxide-based hybrid fillers	84/na	AFX
Zenchroma	President Dental, Germany	2,02E+09	Microhybrid	Universal	UDMA Bis-GMA TEMDMA	Glass powder, silicon dioxide inorganic filler (0.005–3.0 µm).	75/53	ZC
Essentia Universal	GC Corp, Japan	200327A	Microhybrid	Universal	UDMA Bis-MEPP Bis-EMA Bis-GMA TEGDMA	PPF (17 μm): strontium glass (400 nm), lanthanide fluoride (100 nm), fumed silica (16 nm) FAISi glass (850 nm)	81/na	ES

Abbreviations: Bis-GMA = Bisphenol A-glycidyl methacrylate; TEGDMA = triethylene glycol Di methacrylate; UDMA = urethane dimethacrylate; TCD-DI-HEA = 2-propenoicacid; (octahydro-4,7-methano-1H-indene-5-diyl) bis (methyleneiminocarbonyloxy-2,1-ethanediyl) ester; PPF = pre-polymerized filler; SiO_2_ = silicon oxide (silica); ZrO_2_ = zirconium oxide; BaO-Al_2_O_3_-SiO_2_ = Barium aluminosilicate glass; TiO_2_ = Titanium dioxide, YbF_3_ = Ytterbium trifluoride; B_2_O_3_-F-Al_2_O_3_-SiO_2_ = Boro-fluoro-aluminosilicate; CF = Composite filler; Bis-EMA = Ethoxylated bisphenol-A Di methacrylate; Bis-MEPP = Bisphenol-A ethoxylate Di methacrylate; TEMDMA = Tetra-ethylen Di methacrylate; fAlSi = fluoroaluminosilicate; na: not available. * The data were provided by the manufacturers [12,13,14,15,16,17,18].

**Table 2 polymers-14-04987-t002:** Comparison of VHN and DC Values after 24 h and 15 days of the tested materials.

		24 h		15 Days		Test Statistics	*p*
		Mean ± SD	Median (Min-Max)	Mean ± SD	Median (Min-Max)		
Bottom surface	ZC	92.95 ± 6.16	94.7 (86.6–100.8) ^ab^	106.24 ± 19.43	103(90–139.3)	Z = −1.214	0.225
	DO	106.51 ± 19.3	111.3 (79.7–130.7) ^b^	121.44 ± 23.59	116(97.2–153.7)	Z = −1.214	0.225
	VU	103.25 ± 13.66	98.7 (94.5–127.3) ^b^	118.21 ± 31.02	105(96.9–171.7)	Z = −1.483	0.138
	OS	92.3 ± 10.91	94.2 (75.9–104.7) ^ab^	98.77 ± 7.49	101 (90.7–107.7)	Z = −1.214	0.225
	AFX	105.81 ± 8.7	106.5 (97–119.3) ^b^	105.92 ± 12.47	109.3 (88.7–121.7)	Z = −0.135	0.893
	OC	75.82 ± 5.89	74.4 (69.3–82.1) ^ab^	163.87 ± 31.61	177.3 (113–193.3)	Z = −2.023	**0.043**
	ES	46 ± 4.82	46.6 (40.9–52.7) ^a^	113.91 ± 26.61	114.4 (86.9–148.3)	Z = −2.023	**0.043**
	Test statistics	$\chi^{2}=$23.939		$\chi^{2}=$11.909			
	*p*	**0.001**		**0.064**			
Top surface	ZC	127.5 ± 30.15	123 (92.2–174.3)	138.18 ± 28.25	150.7 (98.2–169.7) ^ab^	Z = −0.674	0.500
	DO	141.93 ± 20.85	137.7 (116.3–170.3)	148.87 ± 20.64	146.7 (123.7–172.7) ^ab^	Z = −0.405	0.686
	VU	135.46 ± 27.72	137 (107.7–172)	155.47 ± 25.96	147 (126.3–196) ^ab^	Z = −0.944	0.345
	OS	128.18 ± 24.2	121 (99.9–155)	133 ± 19.26	132.3 (112–162) ^b^	Z = −0.135	0.893
	AFX	138.18 ± 20.36	147.7 (102.2–150)	126.85 ± 26.98	128.4 (97.6–158) ^b^	Z = −0.674	0.500
	OC	114.16 ± 35.85	117.2 (72.9–159.7)	218.93 ± 85.47	184 (143.3–333.7) ^ab^	Z = −2.023	**0.043**
	ES	92.59 ± 24.81	81.3 (68–131.3)	215.33 ± 37.6	205 (176–270.7) ^a^	Z = −2.023	**0.043**
	Test statistics	$\chi^{2}=$9.184		$\chi^{2}$= 16.220			
	*p*	0.164		**0.013**			
Bottom/ Top Ratio (%)	ZC	75.27 ± 13.06	75.1 (57.8–94.1)	81.84 ± 34.29	68.2 (59.7–141.8)	Z = −0.405	0686
	DO	76.3 ± 16.94	82.9 (51.8–94.9)	81.85 ± 13.45	78.6 (69.7–101.5)	Z = −0.405	0.686
	VU	77.93 ± 12.76	84 (58.3–88.7)	75.34 ± 7.6	73 (67.5–87.6)	Z = −0.405	0.686
	OS	75.32 ± 22.28	77.9 (49–104.8)	75.32 ± 10.8	76.3 (63.7–89.7)	Z = −0.135	0.893
	AFX	77.96 ± 12.87	72.1 (64.9–97.3)	86.06 ± 17.45	90.9 (62.9–107.3)	Z = −1.483	0.138
	OC	71.21 ± 19.16	69.8 (46.6–95)	80.46 ± 22.01	75.7 (57.9–108.1)	Z = −1.214	0.225
	ES	52.01 ± 12.45	51.8 (35.5–70.5)	54 ± 13.9	55.8 (32.8–68.6)	Z = −0.135	0.893
	Test statistics	$\chi^{2}$= 7.874		$\chi^{2}=$10.072			
	*p*	0.247		0.122			

Z: Wilcoxon test; $\chi^{2}$: Kruskal–Wallis test. SD: standard deviation; min: minimum; max: maximum. Same letters indicate no statistical difference between the groups.

**Table 3 polymers-14-04987-t003:** Comparison of DC% values by composite type.

Composite Brand	Mean ± SD	Median (Min-Max)	Test Stat.	*p **
OC	52.09 ± 1.71 ^a^	51.38 (50.85–54.03)		
AFX	76.09 ± 4.26 ^ab^	74.11 (73.17–80.98)	23.779	**0.** **001**
OS	64.51 ± 17.78 ^ab^	55.44 (53.10–85.00)		
DO	65.10 ± 1.60 ^b^	64.97 (63.57–66.76)		
EU	68.47 ± 5.99 ^ab^	65.97 (64.13–75.30)		
ZC	54.26 ± 9.92 ^ab^	53.88 (44.54–64.37)		
VU	67.57 ± 0.86 ^b^	67.51 (66.75–68.46)		

* One-way analysis of variance (Welch). Same letters indicate no statistical difference.

**Table 4 polymers-14-04987-t004:** Comparison of F_S_ values according to composite brands and time.

	SS	*D_f_*	MS	F	*p*	*η_p_* ^2^
Composite brand	177696.499	6	29616.083	42.974	**<0.001**	0.672
Time	867.645	1	867.645	1.259	0.264	0.010
Composite brand * time	737.506	6	122.918	0.178	0.982	0.008

F: Analysis of variance test statistics. R^2^: 0.674; Adjusted R^2^: 0.640. SS: sum of squares. *D_f_*: degrees of freedom. MS: mean of squares. *η_p_*^2^: partial eta square.

**Table 5 polymers-14-04987-t005:** Multiple comparison of F_S_ (MPa) values of the tested composites according to time intervals.

Composite Brand	Time		Total
	24 h	15 Days	
OC	82.79 ± 18.59	75.38 ± 20.07	79.08 ± 19.21 ^cd^
AFX	65.34 ± 19.07	55.41 ± 12.6	60.38 ± 16.54 ^d^
OS	82.53 ± 23.79	83.01 ± 29.09	82.77 ± 25.86 ^cd^
DO	142.66 ± 22	137.76 ± 27.23	140.21 ± 24.22 ^ab^
ES	99.59 ± 21.22	98 ± 20.77	98.79 ± 20.45 ^c^
ZC	170.1 ± 46.37	158.25 ± 28.61	164.18 ± 37.99 ^a^
VU	137.37 ± 23.22	137.73 ± 36.75	137.55 ± 29.92 ^b^
**Total**	111.48 ± 44.1	106.5 ± 43.59	108.99 ± 43.76

Same letters indicate no statistical difference.

**Table 6 polymers-14-04987-t006:** Comparison of E_M_ values according to composite brand and time.

	Test Statistics	*p*
Composite brand	9.020	**<0.001**
Time	0.158	0.691
Composite brand * time	1.520	0.958

* Two-way Robust test.

**Table 7 polymers-14-04987-t007:** Multiple comparison of E_M_ (gPa) values of the composites according to time intervals.

Composite Brands	Time		Total
	24 h	15 Days	
OC	2.91 (1.49–4.53)	2.79 (1.91–3.40)	2.87 (1.49–4.53) ^d^
AFX	2.33 (1.63–3.78)	2.18 (1.32–2.78)	2.33 (1.32–3.78) ^e^
OS	3.38 (2.83–8.12)	3.10 (1.95–4.94)	3.32 (1.95–8.12) ^d^
DO	5.52 (3.61–7.04)	4.75 (3.83–6.30)	5.26 (3.61–7.04) ^c^
ES	3.41 (1.41–4.45)	3.28 (2.30–5.43)	3.41 (1.41–5.43) ^ad^
ZC	5.79 (2.23–7.80)	6.45 (4.68–9.06)	6.17 (2.23–9.06) ^b^
VU	4.13 (2.88–6.14)	3.92 (2.22–6.31)	4.01 (2.22–6.31) ^a^
Total	3.53 (1.41–8.12)	3.50 (1.32–9.06)	3.50 (1.32–9.06)

Same letters indicate no statistical difference.

## Data Availability

The data presented in this study are available on request from the corresponding author.

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
