# Peer review of "Assessment of Micro-Hardness, Degree of Conversion, and Flexural Strength for Single-Shade Universal Resin Composites"

_polymers, 2022, doi:10.3390/polym14224987_

Round 1

Reviewer 1 Report

In abstract and conclusion, "clinically acceptable" were mentioned.  But since the evaluation of "clinically acceptance" must be judged from various aspects, it seems inappropriate to mention it in these sections.  And the clinical advantages of these materials should be described in the Discussion section.

Only the samples used for the DC evaluation had their surfaces polished, but the other test samples' surfaces should be polished as well. It is necessary to state why this was not done or why polishing was done only for DC.

The resolution of Figure 1 was quite low and needs to be replaced.

Author Response

Dear Reviewer,

Thank you very much for your time reviewing our manuscript “Assessment of Micro-Hardness, Degree of Conversion, and Flexural Strength for Single-shade Universal Resin Composites”. We very much appreciated your thoughtful and constructive feedback which helped us to improve our analysis and manuscript, further.

We agreed and appreciated all your comments. After considering your comments carefully we reflected them in the manuscript and/or responded them directly through the submission system, copied below. Please kindly note that all changes are made in consultation with all nine co-authors.  We wanted to share t our responses with you. Attached, please find  a point-by-point response to the reviewer’s comments 

It has been a great pleasure working with & learning from you all. Thank you very much again for generously spending your time and sharing your experience & knowledge with us – we already learned a lot from you. After incorporating your comments and suggestions we believe our manuscript has improved substantially and hope it has reached to the standards for all the reviewer’s approval to published.   If you have any further questions or would like to exchange further feedback.

Kind Regards,

Reviewer 2 Report

Title: good

Abstract: - 24h/15 days of what?   Introduction: - Lines 90-93 to the discussion part - Please clarify the originality of the present study   Methods: Lines 114: any sample size test? Lines 144-145: What about 24h and why did the authors not use an artificial saliva?   Results: SEM images should have higher resolution

Author Response

(The authors gave the same response as above.)
